# Associations of body adiposity index, body mass index, waist circumference, and percentage of body fat in young female Emirati adults

**Dalia Haroun❓\*, Maryam Darwish, Aseel Ehsanallah**

Department of Public Health and Nutrition, College of Natural and Health Sciences, Zayed University, Dubai, United Arab Emirates

\* Dalia.haroun@zu.ac.ae

## Abstract

Body Adiposity Index (BAI), which relies on an individual's hip circumference and height, was proposed as an alternative anthropometric measurement to Body Mass Index (BMI). Although this measure has been validated across different populations, its accuracy in predicting percent body fat (%BF) in the United Arab Emirates has not yet been assessed. The objective of this study was to examine the association between BAI, BMI, Waist Circumference (WC), and %BF in young female Emirati adults and determine the relative accuracy of BAI when predicting %BF. A retrospective cross-sectional study was conducted among 95 Emirati women between the ages of 17 and 27. The %BF was measured using the dual-energy X-ray absorption (DXA) scanner. Anthropometric measurements were collected, and BMI and BAI were calculated. BMI and %BF (r = 0.823, p <0.001) showed a greater association than that between BAI and %BF (r = 0.702, p <0.001). A linear regression analysis revealed that BMI was the single best predictor of %BF in the sample ($r^2 = 0.678$, p<0.001). The variation around the regression line for BAI comparisons with %BF (standard error of estimate = 4.879) was greater than BMI comparisons (standard error of estimate = 3.889). BAI was found to significantly underestimate %BF at higher adiposity levels (mean difference = 8.7%). The ROC curve analysis demonstrated that BMI had a higher discriminatory capacity (AUC = 0.891) over WC and BAI. The results demonstrated that BMI was a better predictor of %BF in the sample than BAI and WC. Thus, BMI may be more useful in assessing adiposity in young female Emirati adults than BAI. However, the potential of BAI as an alternative measure of adiposity should continue to be examined.

**Data Availability Statement:** All relevant data are within the manuscript and its Supporting Information files.

## Introduction

Body Mass Index, abbreviated as BMI, is a popular anthropometric measurement that utilizes an individual's weight in kilograms by the square of their height in meters in order to provide an estimate of their body fat percentage. The anthropometric measurement is primarily used

**Funding:** The author(s) received no specific funding for this work.

**Competing interests:** The authors have declared that no competing interests exist.

to evaluate an individual's current state of health in addition to the disease risk associated with an increase in body fat [1]. The BMI measure provides a straightforward and practical way to evaluate individuals for their weight status, such that a higher BMI (BMI $\geq$ 30 kg/m$^2$) is related to obesity, which is linked to a number of comorbid diseases as well as increased mortality [2, 3]. Although it is considered effective for evaluating body composition in clinical field settings, its utility has limitations, which indicates that it is unreliable in predicting body fat percentage due to its inability to differentiate between fat mass (FM) and fat-free mass (FFM) [4]. Furthermore, because obesity is defined as having a BMI that is equal to or greater than 30, the categorization of obesity is subsequently restricted as it overlooks body composition [5].

In addition, the accumulation of visceral fat and adipose tissue, which occurs in obesity, has been linked to a considerable increase in the risk of cardiovascular disease [6]. Therefore, fat distribution is an important factor to take into consideration when assessing an individual's weight status. Complex devices, such as the dual-energy X-ray absorption (DXA) scanner, have been used to accurately assess body composition. DXA evaluates bone mineral content in addition to FM and FFM [7]. Other measurements of body composition include waist circumference (WC), waist-to-hip ratio, and waist-to-height ratio, which are commonly used to assess abdominal obesity [8]. Nonetheless, the DXA technique has established itself as the gold standard for determining an individual's body composition by providing information on both FM and FFM [9]. The utilization of DXA, on the other hand, is prohibitively expensive and only available to a limited population. Because of this, the utilization of modalities that are easier to obtain for evaluating an individual's level of adiposity would have significant practical advantages. Therefore, Bergman et al. proposed a new anthropometric measurement in 2011, known as the body adiposity index (BAI), in order to evaluate body composition by dividing an individual's hip circumference in centimeters by their height in meters [10]. The following formula can be used to calculate BAI:

$$\frac{(hip\ circumferance\ in\ cm)}{(height\ in\ m)^{1.5}} - 18$$

The measure was validated in a population of African American adults before being applied to the Mexican American community [11]. In a clinical setting, BAI may be utilized as a quick and low-cost method for estimating the percentage of body fat.

Furthermore, studies have shown that BAI is a better predictor of the percentage of fat in different populations compared to BMI. For example, a study compared BMI, BAI, and WC with the percentage of fat using the 4-compartment (4C) model in a sample of 188 young adults between the ages of 18 and 30. The results revealed that BAI and WC were more strongly associated with percent fat (r = 0.668 and 0.194, respectively) compared to BMI (r = 0.192, p <0.01) [12]. Similarly, based on measurements obtained from bioelectrical impedance analysis (BIA), it was discovered that BAI (r = 0.74, p < 0.001) is a more precise measurement of body fat than BMI (r = 0.54, p < 0.001) in a sample of 3,200 Caucasian adults [13].

On the other hand, a study has shown that BAI (r = 0.42, p = 0.003) does not correlate significantly with the percentage of fat from DXA compared to BMI (r = 0.65, p = 0.003) in a sample of 19 clinically severe obese women [14]. This may be due to the use of higher BMI levels compared to other studies and the possible overestimation of percent fat from DXA, as errors of overestimation of DXA have been recorded in obese patients [15]. Likewise, a study including 2950 Korean women between the ages of 18 and 39 discovered that BMI (r = 0.935, p < 0.0001) had a stronger correlation with bioelectrical impedance test results for fat mass than BAI (r = 0.735, p < 0.0001) [16]. Moreover, a study compared the association between BAI and percent fat with BMI based on estimations of skinfold thickness, and this study found

that there is a stronger correlation between BMI and body fat (r = 0.839, p < 0.01) than there is between BAI and body fat (r = 0.772, p < 0.01) [17]. The inconsistencies between the results may be due to differences in age, gender, ethnicity, or anthropometric measurements used. Additionally, several studies have concluded that the percentage of body fat discovered by BAI is subject to considerable underestimation or overestimation, particularly in cases where there are lower levels of adiposity [18, 19]. Due to these discrepancies, further research is needed to determine the long-term health implications of these measurements and their utility in a clinical setting.

Moreover, there is a lack of research on the validity of BAI when it comes to estimating the percentage of body fat among the population of the United Arab Emirates (UAE). According to data from the UAE National Health Survey Report of 2017–2018, the prevalence of obesity among women in the UAE is quite high and is estimated to be around 30.6% [20]. This highlights the need for increased public health interventions to address obesity and its associated health risks in the country. Additionally, identifying a more accurate estimation of body composition is of relevance to clinicians with limited access to advanced assessment tools. BAI can serve as a potential alternative to BMI, as it has been found to be a more accurate measurement of body fat in different populations. However, the accuracy of BAI in predicting body fat percentage may be dependent on population characteristics, and the lack of research on its validity in the UAE, especially in women, compels its implementation. In light of this, the aim of this research is to examine the association between BAI, BMI, WC, and percent body fat (%BF) in young female Emirati adults and determine the relative accuracy of BAI when predicting %BF.

## Materials and methods

### Study subjects

In a retrospective analytical cross-sectional study, all data, including questionnaire responses and measurements, were initially collected between March 2016 and March 2017. The current study involved analyzing this comprehensive dataset obtained from a sample of 170 females aged 17 to 27 years The recruited participants included healthy Emirati women attending public universities in the UAE who were chosen via a convenience snowball sampling method. The study was voluntary, and informed written consent was obtained from all those involved. The measurements were taken in the body composition laboratory at Zayed University in Dubai, UAE. The study received full ethical approval from the Zayed University Research Ethics Committee (ZU16_016_F). The selection of participants was based on the following inclusion criteria: female Emiratis between the ages of 17 and 28, fasting for 12 hours, and not menstruating. Exclusion criteria included males, non-Emiratis, those with metabolic disorders (such as diabetes, kidney disease, and hypertension), those using specific medications, pregnant or lactating females, and those who reported weight fluctuations greater than 3 kg. Out of 170 participants, 95 individuals were included in the analysis after accounting for the exclusions. Several channels, including electronic mail, text messages, and word of mouth on campus, were used to spread information about the study and recruit female university students. Students who showed interest in volunteering received either an electronic or paper copy of the information sheet. Students who agreed to participate were then given a consent form to read and sign before the assessment session. The names and contact information of students were stored separately in a locked cabinet, and each student was assigned a unique code to ensure the anonymity and confidentiality of their measurements. It was required that all participants arrive at the lab having fasted for at least 8–12 hours prior to their arrival and with an empty stomach. A checklist was run through to make sure the subjects were eligible for the examination. The consent form and information sheet were then delivered to them for review,

if they hadn't already done so. The session lasted between 60 and 90 minutes, during which data and measurements were collected.

## Data collection

**Questionnaire.** A structured questionnaire was administered by a qualified research assistant to collect personal information such as age, sex, ethnicity, marital status, body weight, height, medical history, and medications used, if applicable. Additionally, information on weight fluctuations was gathered by asking participants, "Over the last 3 months, have you experienced any changes in your weight? If so, how many kilograms?"

**Anthropometric measurements and body fat analysis.** Body weight was measured to the nearest 0.01 kg using the BodPod System electronic scale (Body Composition System; Life Measurement, Incorporated, Concord, CA). Height was measured to the nearest 0.1 cm using a wall-mounted stadiometer (TANITA HR-200, China). The waist circumference was measured at the narrowest point to the nearest 0.1 cm over light clothing using a non-elastic, flexible tape (SECA 201) [21]. The hip circumference was measured at the widest portion of the hip [21]. BMI was calculated by dividing the weight in kilograms by the square of the height in meters ($kg/m^2$). The classification of overweight is defined by the World Health Organization (WHO) as a BMI between 25 and 29.9 $kg/m^2$, and the classification of obesity is defined as a BMI greater than 30 $kg/m^2$ [22]. BAI was calculated as [(hip circumference (cm) / height $(m)^{1.5}$)– 18]. %BF was estimated by the Lunar Prodigy iDXA scanner from GE Healthcare, United States. The DXA scanner provides accurate data on bone and soft tissue composition, including bone mineral density (BMD), lean and fat tissue mass, and fat percentage. Whole-body scans were performed on participants who were instructed to lie flat on the scanning bed with their hands at their sides and to refrain from moving throughout the measurement.

## Statistical analysis

The Statistical Package for the Social Sciences (SPSS), Version 29, was used for statistical analysis. The quantitative variables are expressed as means ± standard deviations (SD). Two-tailed p values <0.05 were deemed significant. All data were tested for their normal distribution using the Kolmogorov-Smirnov test. While WC, BAI, and BMI did not conform to a normal distribution, their distributions were approximately symmetric. The %BF assessed by DXA was used as the 'gold standard' to determine %BF. Pearson correlations were applied to assess the associations between BMI, BAI, WC, and %BF. Paired sample t-tests were used to test differences in mean %BF between DXA and BAI in the sample overall and within different BMI categories as well as different adiposity levels. Bland and Altman analysis was used to assess the agreement between BAI and %BF (DXA) to evaluate body adiposity [23]. The diagnostic accuracy of BMI, BAI, and WC were assessed by constructing ROC curves to detect %BF. Cut-off values were calculated based on the point on the RIC curve with the lowest value for the formula $(1-sensitivity)^2 + (1-specificity)^2$. In addition, linear regression was used to evaluate the best predictors of %BF when at least one predictor was significant. The standard error of the estimate (SEE), defined as the standard deviation of the data points around the regression line, and the coefficient of determination ($R^2$) were computed.

## Results

### Description of study sample characteristics

Descriptive characteristics for all subjects included in the analysis are shown in Table 1. The sample included a total of 95 participants. The mean age of the subjects was 19.7 years. The

**Table 1.  Descriptive characteristics of the study sample (n = 95).**

|  | Minimum | Maximum | Mean ± SD |
|---|---|---|---|
| Age (years) | 17 | 28 | 19.7 ± 2 |
| Height (cm) | 139.5 | 180 | 158.4 ± 6.4 |
| Weight (kg) | 36.1 | 113.8 | 58 ± 15.4 |
| BMI (kg/m$^2$) | 14.8 | 38.9 | 22.9 ± 5.2 |
| WC (cm) | 52 | 105 | 68.7 ± 10.8 |
| BAI | 11.7 | 52.4 | 30.4 ± 5.6 |
| % Fat (DXA) | 23.9 | 56.2 | 39.1 ± 6.8 |

mean BMI was 22.9 kg/m$^2$ (14.8 to 38.9 kg/m$^2$), and the mean BAI was 30.4 (11.7 to 52.4). Among 95 subjects, 19 (20%) were found to be underweight, 52 (54.7%) were of normal weight, 14 (14.7%) were overweight, and 10 (10.5%) were obese. The mean waist circumference was 68.7 cm, and the mean %BF was 39.1%. The number of participants who had an increased risk of metabolic diseases was 6 (6.3%), classified by the WHO as a waist circumference >88 cm [21].

## Correlations between body mass index, body adiposity index, waist circumference, and percentage of body fat

BMI, BAI, and WC were all significantly correlated with %BF in the sample. The correlation between BMI and %BF (r = 0.823, p < 0.001) was found to be much greater than that between BAI and %BF (r = 0.702, p < 0.001). The correlation between WC and %BF (r = 0.766, p < 0.001) was also stronger than that between BAI and %BF. Strong correlations were also found between BMI and WC (r = 0.903, p < 0.001) and between BMI and BAI (r = 0.837, p < 0.001). A correlation matrix is displayed in Table 2. When separating individuals into different BMI weight status classifications, the correlation between BMI and %BF, WC and %BF, and BAI and %BF in overweight/obese individuals (r = 0.769, 0.621, and 0.741, respectively, all p < 0.001) was greater than that in underweight/normal weight individuals (r = 0.687, 0.605, and 0.369, respectively, all p < 0.001). When individuals were separated according to adiposity levels, only significant correlations were found amongst those >20% BF. The correlations of the measures in different BMI categories and different levels of adiposity are presented in Table 3.

## Body mass index, body adiposity index, and waist circumference as predictors of percent body fat

A clustered bar graph was used to illustrate the accuracy of classifying adiposity levels by BMI categories versus BAI. The graph revealed that BAI tends to underestimate individuals with

**Table 2.  Correlation matrix of body mass index, waist circumference, body adiposity index, and percentage of fat (n = 95).**

|  | BMI | WC | BAI | Fat (%) |
|---|---|---|---|---|
| BMI | 1 | 0.903** | 0.837** | 0.823** |
| WC | 0.903** | 1 | 0.665** | 0.766** |
| BAI | 0.837** | 0.665** | 1 | 0.702** |
| Fat (%) | 0.823** | 0.766** | 0.702** | 1 |

*BMI*, Body Mass Index. *WC*, waist circumference. *BAI*, Body Adiposity Index. **. Correlation is significant at the 0.001 level (2-tailed).

**Table 3. Pearson's correlation coefficients between BMI, WC, BAI, and %BF (measured by DXA) in different BMI categories and different levels of adiposity.**

| | | n | BMI | WC | BAI |
|---|---|---|---|---|---|
| **BMI Categories** | Underweight-normal weight | 71 | 0.687** | 0.605** | 0.369** |
| | Overweight-obese | 24 | 0.769** | 0.621** | 0.741** |
| **Level of adiposity (%BF by DXA)** | 20–30% | 10 | - 0.248 | 0.363 | - 0.288 |
| | 30–40% | 47 | 0.434** | 0.395** | 0.370* |
| | > 40% | 38 | 0.776** | 0.643** | 0.740** |

*BMI*, Body Mass Index. *WC*, waist circumference. *BAI*, Body Adiposity Index. %BF, % body fat. **. Correlation is significant at the 0.001 level (2-tailed).

higher adiposity levels compared to BMI (Fig 1). The sample was divided into different adiposity levels according to DXA %BF measurements.

Paired sample t-test in the whole sample showed that BAI significantly underestimated % BF compared to DXA (8.7 ± 4.9%; p<0.001). BAI was also shown to significantly underestimate %BF in the underweight/normal weight (8.4 ± 5.3%; p<0.001) as well as the overweight/ obese (9.6 ± 3.6%; p<0.001) BMI categories.

Individuals were then divided according to %BF (DXA) as shown in Fig 2. BAI was found to significantly underestimate %BF at higher adiposity levels ($\geq$ 30%). No significant difference was found among those with less than 30% fat.

The Bland-Altman plot (Fig 3) assessed the degree of individual agreement between differences between BAI and %BF assessed by DXA and their means. Compared to the reference method to estimate adiposity (DXA), BAI was found to underestimate % BF (mean difference = 8.7%, ranging from 0.9–18.3%).

Fig 4 illustrates a linear regression analysis exploring the association between BMI, BAI, WC, and %BF by DXA). The regression lines assessed the correlation and agreement between BMI, BAI, and WC with % BF estimates from DXA.The linear regression analysis showed that BMI was the best predictor of % BF ($R^2$ = 0.678), compared with BAI ($R^2$ = 0.493) and WC ($R^2$ = 0.587). Moreover, the standard error of the estimate (SEE), also known as the variation around the regression line for BAI comparisons with %BF (SEE = 4.88) was greater than BMI (SEE = 3.89) and WC comparisons (SEE = 4.40).

The area under the ROC curve to detect excess % BF ($\geq$35%) for BMI was higher than for BAI and WC (Fig 5). The ROC curve showed that the cut-off point for BMI was 20.9 and provided a sensitivity of 80.6% and specificity of 89.3% for the diagnosis of obesity based on %BF ($\geq$35%). The cut-off point for BAI was 29.1 and provided a sensitivity of 77.6% and specificity of 78.6%. For WC, the cut-off point was 63.6 cm and provided a sensitivity of 81% and specificity of 78.6%.

## Discussion

The aim of this study was to examine the association between BAI, BMI, WC, and %BF and determine the relative accuracy of BAI when predicting %BF. BAI, which was proposed to be a direct estimate of body adiposity, should be validated in various populations. To the best of our knowledge, this is the first study that investigates the validity of this measure in a population of Emirati women. The significance of this issue lies in the escalating prevalence of overweight/obesity and metabolic disorders within the population of the UAE, especially in women [20]. Therefore, it is vital to develop a cost-effective and precise method for assessing body fat in this population.

In a sample of 95 young Emirati women (17–28 years), the results of this analysis indicate that BMI and WC are more strongly associated with %BF obtained from DXA when compared

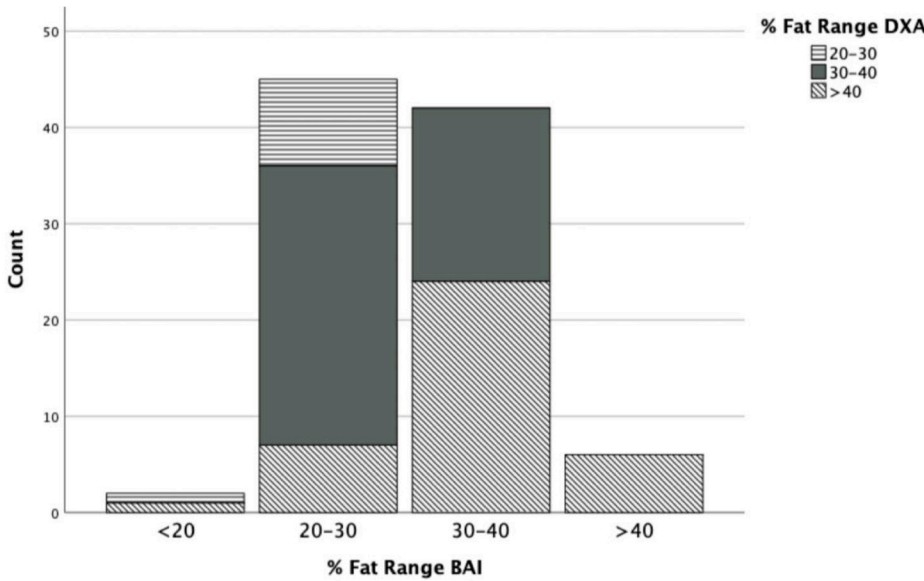

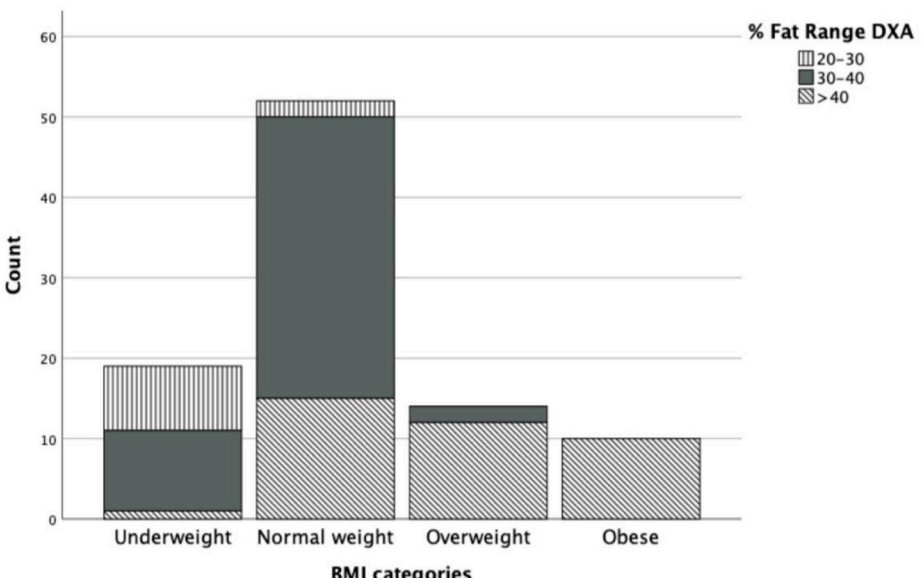

**Fig 1.** Classification of adiposity levels measured by % BF by DXA and according to BMI status (top) versus classification of adiposity using BAI (bottom).

to BAI. This result differs from previous research efforts in which varying measurement tools were used to collect percent body fat, encompassing a wide array of population groups [12, 13]. One study found that in a sample of 188 racially diverse young adults, including Caucasian, African American, and Hispanic individuals, between the ages of 18 and 30 years, BAI and WC were more strongly associated with %BF as measured by the 4C model compared to BMI (r = 0.668 and 0.194, respectively, vs. 0.192, p < 0.01) [12]. Similarly, previous research involving 3,200 Caucasian adults between the ages of 18 and 65 years in Spain showed that

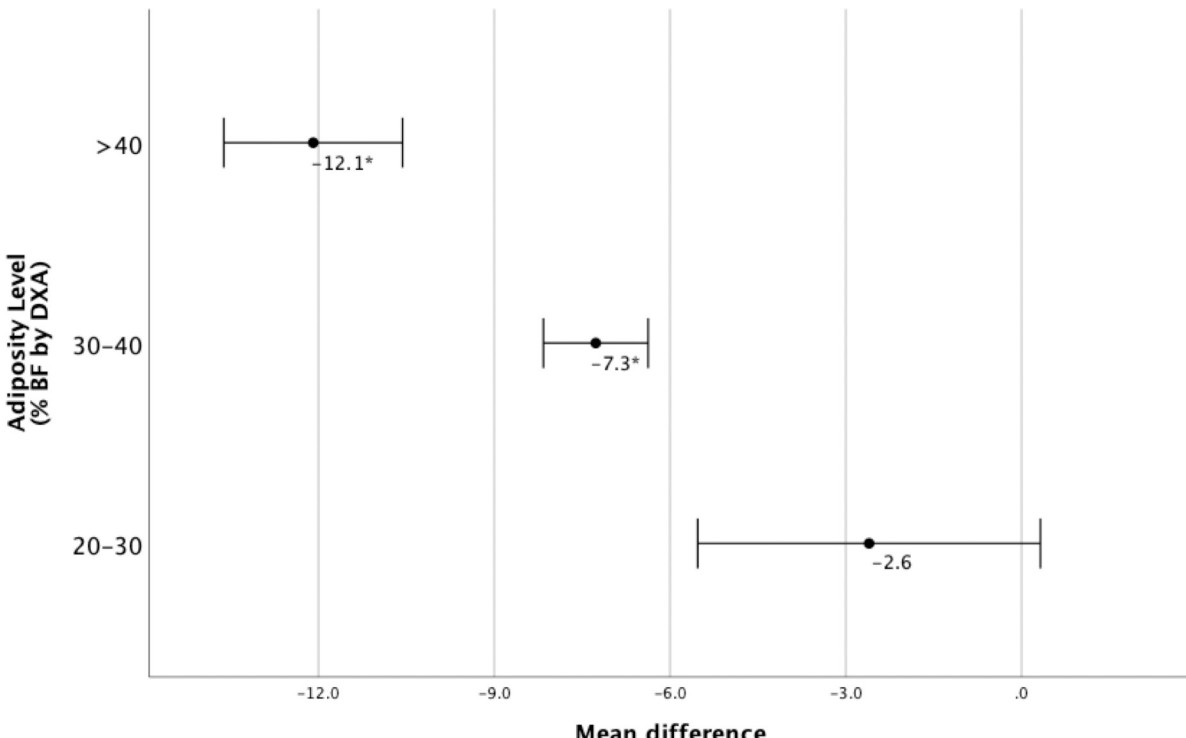

**Fig 2. Comparison of %BF measurements using DXA and BAI across varying levels of adiposity.** Mean difference = BAI-% BF by DXA.
*p < 0.001.

BAI is a more precise measurement of body fat as measured by BIA than BMI (r = 0.74 vs. 0.54, p < 0.001) [13]. The conflicting results could be due to differences in sample characteristics and measurement techniques. Both studies used a larger sample size compared to this study, which could have contributed to the opposing results. A larger sample size may provide higher statistical power to detect significant associations, resulting in different conclusions regarding the strength of associations between BMI, WC, BAI, and %BF. Furthermore, the studies have used different measurement techniques to assess body composition. For example, in the first study, the 4C model, which combines measurements from DXA, bioimpedance spectroscopy, and underwater weighing, was used to estimate body fat [12], and in the second study, BIA was used to measure %BF [13]. Consequently, the discordant results could be the result of variations in the accuracy, precision, and validity of these measurement techniques.

In contrast, based on data from a sample of 19 women with clinically severe obesity with an average age of 32 years from various ethnic backgrounds, including Caucasian, African American, and Hispanic individuals, BAI does not significantly correlate with %BF obtained from DXA compared to BMI (r = 0.42 vs. 0.65, p = 0.003) [14]. This is in agreement with the results of this study as well as those of other studies that found a greater association between BMI and %BF as opposed to BAI [16, 17]. For instance, one study found that in a sample of 2950 Korean women between the ages of 18 and 39, BMI (r = 0.935, p < 0.0001) had a stronger correlation with fat mass obtained from bioelectrical impedance tests compared to BAI (r = 0.735, p < 0.0001) [16]. Possible explanations for the similarity between the studies mentioned in showing a stronger relationship between BMI and %BF than BAI include similar characteristics of the study populations. Both studies that were mentioned collected measurements from women, which could have influenced the associations between body composition measures

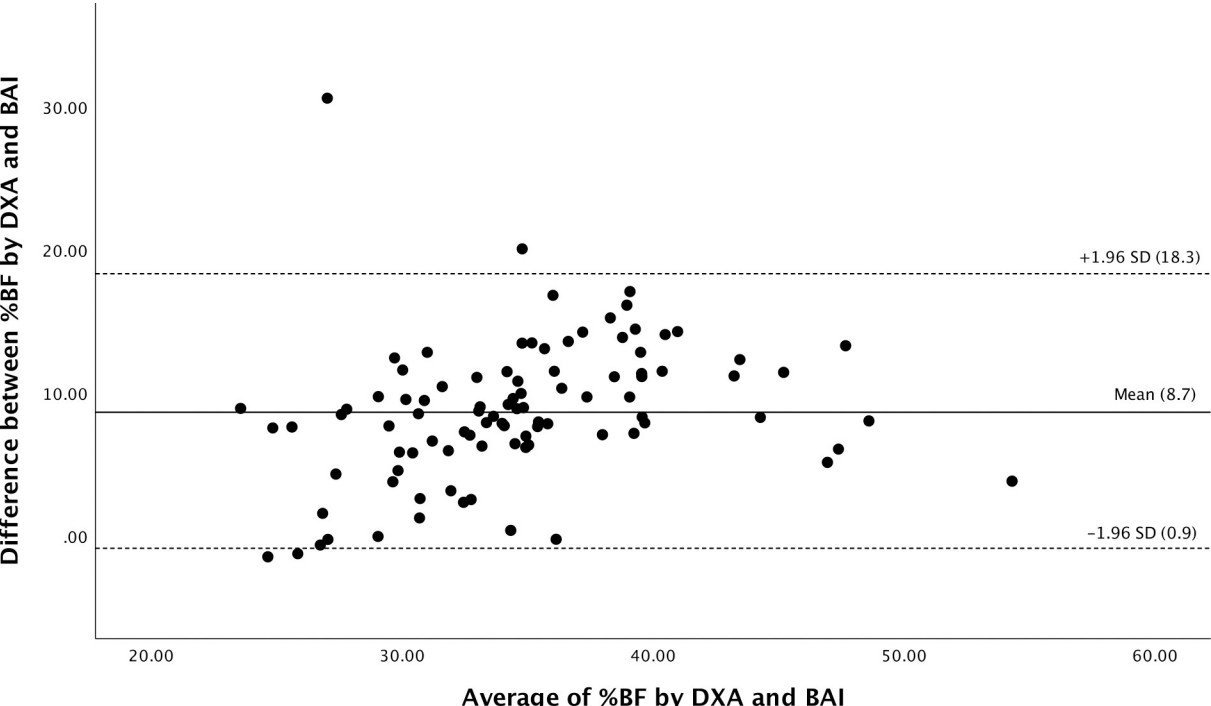

**Fig 3. Bland and Altman's limits of agreement plot of % BF (DXA) and BAI.** The differences between the methods are plotted against the average of them. The solid line represents the mean bias between BAI-%BF (DXA) and the dashed lines indicate mean ±1.96 SD (95% confidence interval of limits of agreement).

and %BF. In addition, in the study that included women with clinically severe obesity, higher BMI levels were obtained, which may have affected the association between BAI and %BF. It has been reported that BAI is less accurate in obese individuals due to the assumption that BAI is unaffected by the distribution of adipose tissue [11]. Thus, in individuals with obesity, BMI may be more indicative of overall adiposity. Additionally, DXA may be less accurate in individuals with severe obesity, as it has been found to overestimate %BF in these individuals [15].

In addition, the correlations between BAI, BMI, WC, and %BF in this study were found to be stronger in overweight/obese individuals compared to underweight/normal weight individuals. This result is similar to the findings of another study that found a greater correlation between BMI and %BF in young women with obesity compared to those who were underweight [24]. This indicates that as BMI increases in the sample, %BF tends to increase to a greater extent in women with obesity than in women considered to be underweight. This may be explained by the fact that in individuals with obesity, a greater proportion of body weight is made up of fat, whereas in individuals who are underweight, a greater proportion is made up of lean body mass.

In this study, the Bland-Altman plot showed that BAI consistently underestimated %BF compared to DXA. The extent of this underestimation varied among individuals, indicating that BAI may not be as accurate as DXA for estimating %BF, especially in individuals with higher levels of adiposity. Confirming the findings of this study, Johnson et al. revealed in a study of 623 individuals, 332 of whom were women, that BAI underestimated %BF in females and individuals with higher levels of adiposity [18]. Additionally, Geliebter et al. found in a study of 19 women with severe obesity that BAI underestimated the percentage of body fat (%BF) [14]. However, comparing the results of this study with those of previous research is challenging due to the difference in devices used, such as BIA.

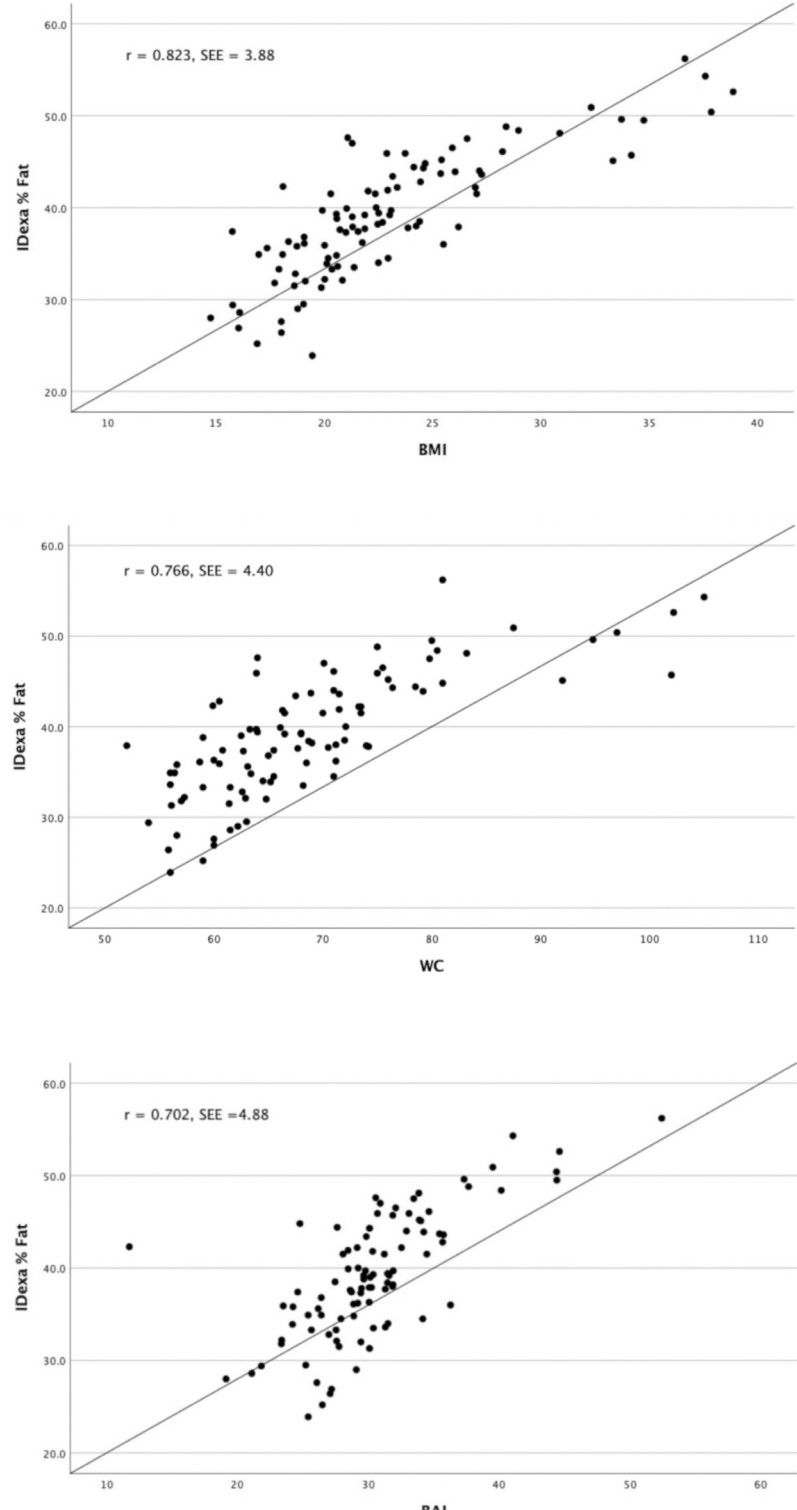

**Fig 4. Regression between body mass index (BMI) / body adiposity index (BAI) and percent body fat (%BF).** (A) BMI and %BF. (B) WC and % BF (C) BAI and %BF.

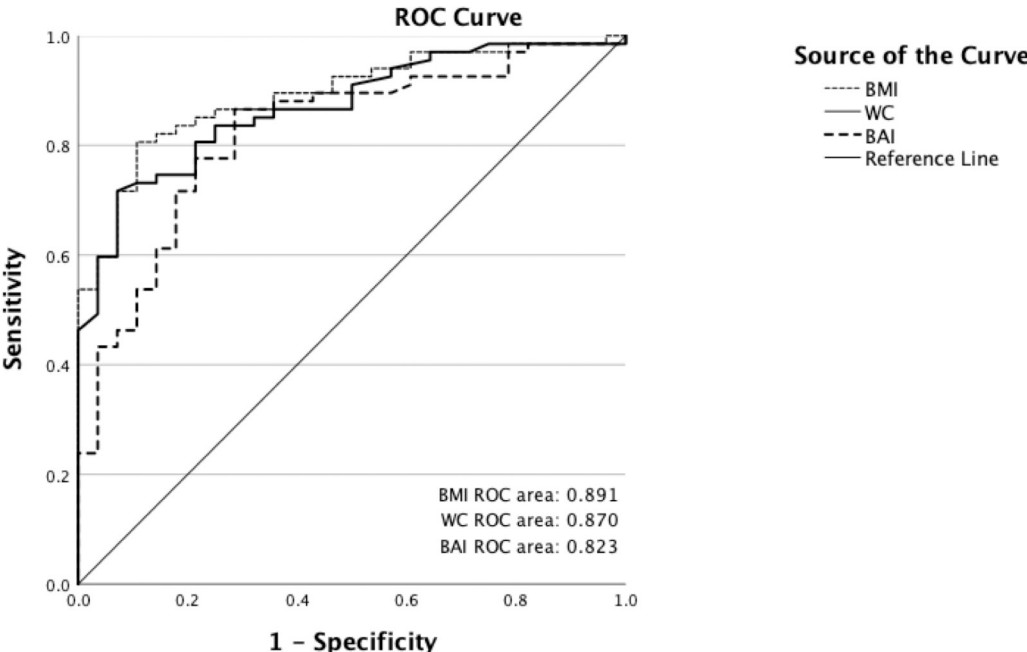

**Fig 5. ROC analysis for sensitivity and specificity of BMI, WC, and BAI in detecting obesity by %BF (assessed by DXA).** The reference line is represented by the solid line.

Similarly, the results from the paired sample t-test revealed that the BAI significantly underestimated %BF when compared to DXA. This underestimation was consistent across different BMI categories. This was in accordance with a study by Ramírez-Vélez et al. on a sample of 96 individuals, which found that BAI underestimated %BF at all levels of adiposity.

Moreover, linear regression analysis revealed that BMI was the greatest predictor of %BF from DXA, indicating that BMI is a reliable tool for evaluating body fat. The variation (SEE) around the regression line for BAI comparisons with %BF was greater than BMI comparisons. In other words, the variability of predicted %BF values around the regression line is larger when using BAI as opposed to BMI. This indicates that in predicting %BF, the predictions based on BAI are less accurate than those based on BMI. This is in agreement with prior studies that reported BMI as the single best predictor of %BF from BAI [14] and skinfold thickness [17].

Furthermore, the linear regression analysis showed that BMI was significantly associated with %BF, whereas the association between BAI/WC and %BF was insignificant. An explanation for this could be the presence of collinearity between the variables, which can make it hard to detect significant associations between independent variables and the outcome variable. This is evidenced by Pearson's correlations, which indicate significant associations between the independent variables, suggesting the existence of collinearity. To address this issue, eliminating one of the collinear variables, combining them into a single variable, or employing a modeling technique that is less sensitive to collinearity can improve the stability and reliability of the regression estimates.

The ROC curve analysis demonstrated that BMI had a higher discriminatory capacity (AUC = 0.891) over WC and BAI in distinguishing obesity based on %BF ($\geq$ 35%). These results were consistent with other studies that suggest BMI be a more reliable indicator for identifying individuals with obesity based on %BF [13, 14, 16].

Moreover, it is important to note that although this type of research provides a chance to establish associations between BAI, BMI, WC, and %BF, it is still challenging to decide when to employ different anthropometric indices because there doesn't seem to be a clear advantage to using one measure over another. As adiposity can vary according to age, race, ethnicity, level of activity, and among individuals with comparable BMIs, it may not be wise to conclude that a single body composition measure is appropriate for all individuals alike. Overall, this study provides evidence that BMI may be better at predicting %BF obtained by DXA than BAI in young Emirati women; however, obtaining a modeling technique to eliminate collinearity could lead to higher precision in the results. Additionally, as this was a cross-sectional study, it is uncertain whether BMI or BAI could be more accurate in detecting changes in %BF. Future research might benefit from a prospective analysis of the relative accuracy of these anthropometric measurements in predicting changes in %BF, which was outside the domain of the current analysis.

The limitations of this study are thoroughly acknowledged. Firstly, the study used convenience sampling, which may have introduced selection bias into the study. The participants were recruited based on their availability and willingness to participate, which may not be representative of the entire population of young female Emirati adults. Therefore, the results may not be generalizable to the entire population. Secondly, there were only 95 participants in the sample, which may have limited the statistical power and precision of the study. With an insufficient sample size, the risk of random error and the inability to detect significant associations between variables may be increased. Thirdly, the study had collinearity issues between the variables which may have contributed to the lack of significance of the observed associations. Collinearity occurs when independent variables are highly correlated, making it difficult to determine the true effects of each variable.

Furthermore, the study included only young female Emirati adults, which may limit the applicability of the findings to other age groups or genders. In addition, only a limited number of variables related to adiposity were assessed, and other factors such as genetics, lifestyle behaviors, and socioeconomic status were not taken into consideration. By omitting these variables, the study may have overlooked important confounding variables that may have affected the associations between BMI, BAI, WC, and %BF. This is because genetics, lifestyle choices, and socioeconomic status play a big role in adiposity and %BF. If these factors aren't taken into account in the study, it could limit the ability to accurately compare the accuracy of BMI, BAI, and WC in predicting %BF. Lastly, as this was a retrospective cross-sectional study, the precision of BAI, WC, and BMI in monitoring changes in %BF is unclear. Future longitudinal studies may benefit from a prospective analysis of the accuracy of these body composition measures in detecting changes in %BF.

## Conclusion

This study provides important insights into the associations between BAI, BMI, WC, and %BF in young female Emirati adults. The study found that BMI was a better predictor of %BF in the sample than BAI and WC. The correlation between BMI and %BF was stronger than the correlation between BAI and %BF. These findings have important implications for the assessment of adiposity in young female Emirati adults. Based on these results, BMI may be used as a reliable measure of adiposity and in predicting %BF, particularly in populations with characteristics that are comparable to those of the sample of this study. This recommendation aligns with existing literature on the subject [14, 17]. Nonetheless, the results of this study should be treated with caution due to the limitations previously mentioned. Consequently, it is recommended that future research develop a specialized statistical model and include the influence

of genetics, lifestyle behaviors, and socioeconomic status on %BF in the analysis of a similar population in order to obtain a more complete understanding of the factors that influence these anthropometric measurements.

Furthermore, the results emphasize the importance of considering a person's weight status when selecting an appropriate measure of adiposity. According to the results of the study, individuals who were overweight or obese had a better correlation between their BMI, WC, BAI, and %BF than individuals who were underweight or of normal weight. This suggests that these measures may be more accurate for those with higher levels of adiposity. Future research should focus on investigating the efficacy of these metrics in various populations in order to ascertain their applicability across settings. Moreover, although BAI did not demonstrate a significant association with %BF as BMI did, it may still have some importance as a marker of adiposity in certain populations. As obesity is a growing concern in the UAE, a better understanding of accurate measures of adiposity is essential to developing effective prevention and management strategies. Therefore, the potential of BAI as an alternative measure of adiposity should continue to be examined, particularly in groups where established measures such as BMI may not be as successful.

## Supporting information

**S1 Data.**
(SAV)

## Acknowledgments

The authors would like to acknowledge all the participants who have provided their tremendous support during the data collection stage. We appreciate the help of the lab personnel at Zayed University and the participants' extreme patience and time generosity during data collection. We also thank Dr. Rafiq Hijazi for his input and advice in relation to statistical analysis.

## Author Contributions

**Formal analysis:** Dalia Haroun, Maryam Darwish.

**Project administration:** Dalia Haroun.

**Supervision:** Dalia Haroun.

**Writing – review & editing:** Dalia Haroun, Maryam Darwish, Aseel Ehsanallah.

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
