## [Decision Letter · Decision Letter 0]

6 Feb 2024

PONE-D-23-35965Associations of body adiposity index, body mass index, waist circumference, and percentage of body fat in young female Emirati adultsPLOS ONE

Dear Dr. Haroun,

Thank you for submitting your manuscript to PLOS ONE. After careful consideration, we feel that it has merit but does not fully meet PLOS ONE’s publication criteria as it currently stands. Therefore, we invite you to submit a revised version of the manuscript that addresses the points raised during the review process.

 Please address all the comments raised by the reviewersThis manuscript requires a major revision==============================

We look forward to receiving your revised manuscript.

Kind regards,

Fredirick Lazaro mashili, MD, PhD

Academic Editor

PLOS ONE

Journal Requirements:

Additional Editor Comments:

Please address all the concerns raised by both the reviewers

Reviewers' comments:

Reviewer's Responses to Questions

**Comments to the Author**

1. Is the manuscript technically sound, and do the data support the conclusions?

Reviewer #1: Partly

Reviewer #2: Yes

2. Has the statistical analysis been performed appropriately and rigorously? 

Reviewer #1: I Don't Know

Reviewer #2: Yes

3. Have the authors made all data underlying the findings in their manuscript fully available?

Reviewer #1: Yes

Reviewer #2: Yes

4. Is the manuscript presented in an intelligible fashion and written in standard English?

Reviewer #1: Yes

Reviewer #2: Yes

5. Review Comments to the Author

Reviewer #1: While validating the use of BAI in estimating % body fat among Emirati women is novel, given the small sample size and heterogeneity of the population, concluding BMI is superior to BAI in predicting %BF might not be scientifically sound. With this sample size, the population must be tightly homogeneous to draw such conclusion ( eg only obese women, only underweight women etc etc). Since this information is important and could inform future large sample size studies, the following major revisions would improve the quality of the manuscript.

1. The authors should try to make this manuscript more descriptive. Eg portraying histograms or bar charts with proportions of those with high BMI or BAI classifying them to have high body fat, who also have high body fat based on DEXA. Or correlation and regression figures for various BMI categories or any way just to make the manuscript more descriptive. This will help to suggest and not to conclude that BMI is likely superior to BAI pending future research with larger sample sizes or homogeneous populations. Would be good to have a manuscript that has looked at different angles of the 95 participants to convince that even though the sample size is small, it definitely suggest that BMI is more superior than BAI.

Reviewer #2: Comments on the paper entitled “Associations of body adiposity index, body mass index, waist circumference, and percentage of body fat in young female Emirati adults”

1.Line 50, I suggest rewriting the sentence to read, Body Mass Index abbreviated as BMI, and not Body Mass Index also known as BMI

2.Line 150, How did you measure weight fluctuations

3.Authors should mention the Names, Brands, Models, and Cities of manufacturers for all types of equipment and tools that were used in measuring all variables

4.It is stated that this study included healthy Emirati women attending “public universities” in the UAE who were chosen via a convenience sampling method (lines 123-124). It is also well explained that” The measurements were taken in the body composition laboratory at Zayed University in Dubai, UAE”. The clarifications are needed to disclose how participants from other universities were conveyed to the measurement center.

5.State if any statistical test for the normality of the data was carried out before running regression and correlation analysis

6.line 292 what is BIA?

7.One of the aims of this study was to determine the relative accuracy of BAI when predicting %BF, However, this objective is not clearly shown through the subsequent sections of the manuscript, in other words the objective should be featured in data collection, analysis, and discussion.

6. PLOS authors have the option to publish the peer review history of their article (what does this mean?). If published, this will include your full peer review and any attached files.

Reviewer #1: **Yes: **Fredirick Lazaro Mashili

Reviewer #2: **Yes: **Oscar Mbembela

---

## [Author Response · Author response to Decision Letter 0]

4 Mar 2024

Editor Comments

Response: We have checked that the manuscript meets PLOS one’s style requirements.

Response: informed written consent was collected from all participants included in the study.

This has been highlighted in the Material and Methods section.

Reviewer Comments

Reviewer #1: While validating the use of BAI in estimating % body fat among Emirati women is novel, given the small sample size and heterogeneity of the population, concluding BMI is superior to BAI in predicting %BF might not be scientifically sound. With this sample size, the population must be tightly homogeneous to draw such conclusion ( eg only obese women, only underweight women etc.). Since this information is important and could inform future large sample size studies, the following major revisions would improve the quality of the manuscript.

1. The authors should try to make this manuscript more descriptive. Eg portraying histograms or bar charts with proportions of those with high BMI or BAI classifying them to have high body fat, who also have high body fat based on DEXA. Or correlation and regression figures for various BMI categories or any way just to make the manuscript more descriptive. This will help to suggest and not to conclude that BMI is likely superior to BAI pending future research with larger sample sizes or homogeneous populations. Would be good to have a manuscript that has looked at different angles of the 95 participants to convince that even though the sample size is small, it definitely suggest that BMI is more superior than BAI.

Response: We have analysed the data further taking into account the reviewer comments. Our sample was first divided according to % fat levels (by DXA). Then using a using a clustered bar graph, the accuracy classifying adiposity using BMI categories versus BAI were shown. (Fig 1). Furthermore, ROC analysis was used to investigate which measure was better able to detect excess % BF (using a cut-off of 35%) (Fig 4). The results all revealed that BMI was more superior than BAI.

Reviewer #2: Comments on the paper entitled “Associations of body adiposity index, body mass index, waist circumference, and percentage of body fat in young female Emirati adults”

1.Line 50, I suggest rewriting the sentence to read, Body Mass Index abbreviated as BMI, and not Body Mass Index also known as BMI

Response: This has been revised and reads: “Body Mass Index abbreviated as BMI”.

2.Line 150, How did you measure weight fluctuations

Response: Weight fluctuations was gathered by asking participants, "Over the last 3 months, have you experienced any changes in your weight? If so, how many kilograms?". Further clarification has been added to the manuscript under the data collection section.

3.Authors should mention the Names, Brands, Models, and Cities of manufacturers for all types of equipment and tools that were used in measuring all variables

Response: This information has been added in the manuscript, material and methods section.

4.It is stated that this study included healthy Emirati women attending “public universities” in the UAE who were chosen via a convenience sampling method (lines 123-124). It is also well explained that” The measurements were taken in the body composition laboratory at Zayed University in Dubai, UAE”. The clarifications are needed to disclose how participants from other universities were conveyed to the measurement center.

Response: This was done via snowball sampling. It is now added to the manuscript.

5.State if any statistical test for the normality of the data was carried out before running regression and correlation analysis

Response: Normality was assessed using the using the Kolmogorov-Smirnov test- information has been added to the manuscript. % fat was normally distributed; however, WC, BAI and BMI were not. However, the distributions were approximately symmetric with one unusual point. As evidence of robustness, the removal of this point did not change the reported results. Therefore, this should not impact the statistical validity of the correlation and regression analyses.

6.line 292 what is BIA?

Response: This was a typing error and should have read BAI. It has been fixed.

7.One of the aims of this study was to determine the relative accuracy of BAI when predicting %BF, However, this objective is not clearly shown through the subsequent sections of the manuscript, in other words the objective should be featured in data collection, analysis, and discussion.

Response: Further analyses were carried out to test this aim, and was incorporated into the various sections of the manuscript. These include: (a) Paired sample t-tests for the sample overall as well as across various BMI categories and across different adiposity levels were used to test differences in % BF between DXA and BAI (Table 4), (b) Bland-Altman analysis was used to test individual agreement between BAI and % BF (DXA) (Figure 2) , (c) Linear regression analysis was carried out to test the best predictor of % fat (Table 5) (d) ROC analysis for sensitivity, specificity of BAI in detecting obesity by % BF (assessed by DXA) (Figure 4).

---

## [Decision Letter · Decision Letter 1]

3 Apr 2024

PONE-D-23-35965R1Associations of body adiposity index, body mass index, waist circumference, and percentage of body fat in young female Emirati adultsPLOS ONE

Dear Dr. Haroun,

Thank you for submitting your manuscript to PLOS ONE. After careful consideration, we feel that it has merit but does not fully meet PLOS ONE’s publication criteria as it currently stands. Therefore, we invite you to submit a revised version of the manuscript that addresses the points raised during the review process.A minor revision is required for this manuscriptPlease address all the comments thoroughly and accordinglyThe suggested minor revision will make the manuscript clearer, well presented and better aligning to journal's style and formatting ==============================

We look forward to receiving your revised manuscript.

Kind regards,

Fredirick Lazaro mashili, MD, PhD

Academic Editor

PLOS ONE

Journal Requirements:

Additional Editor Comments:

Upon further review, minor revisions have been recommended to enhance the manuscript's clarity and presentation. Specifically, it has been suggested that the current repetitive use of tables for presenting results could be diversified to improve readability and simplicity. Implementing these changes will further strengthen your manuscript.

Reviewers' comments:

Reviewer's Responses to Questions

**Comments to the Author**

1. If the authors have adequately addressed your comments raised in a previous round of review and you feel that this manuscript is now acceptable for publication, you may indicate that here to bypass the “Comments to the Author” section, enter your conflict of interest statement in the “Confidential to Editor” section, and submit your "Accept" recommendation.

Reviewer #1: All comments have been addressed

Reviewer #2: All comments have been addressed

2. Is the manuscript technically sound, and do the data support the conclusions?

Reviewer #1: Partly

Reviewer #2: Yes

3. Has the statistical analysis been performed appropriately and rigorously? 

Reviewer #1: Yes

Reviewer #2: Yes

4. Have the authors made all data underlying the findings in their manuscript fully available?

Reviewer #1: Yes

Reviewer #2: Yes

5. Is the manuscript presented in an intelligible fashion and written in standard English?

Reviewer #1: Yes

Reviewer #2: Yes

6. Review Comments to the Author

Reviewer #1: While most comments have been addressed, some require further attention. Given the descriptive nature of the manuscript, avoiding the repetitive use of tables for presenting results could enhance clarity and simplicity. Diversifying the methods of data presentation beyond just tables would improve the manuscript.

Reviewer #2: I have gone through the author responses to the raised comments, and the comments were satisfactorily responded

7. PLOS authors have the option to publish the peer review history of their article (what does this mean?). If published, this will include your full peer review and any attached files.

Reviewer #1: **Yes: **Fredirick mashili

Reviewer #2: **Yes: **Oscar Mbembela

---

## [Author Response · Author response to Decision Letter 1]

5 Apr 2024

Upon further review, minor revisions have been recommended to enhance the manuscript's clarity and presentation. Specifically, it has been suggested that the current repetitive use of tables for presenting results could be diversified to improve readability and simplicity. Implementing these changes will further strengthen your manuscript.

Response: We thank the editor and reviewer for their comments, and have revised the presentation of results in two tables in the manuscript (1 table was converted into a figure and the other deleted) to enhance clarity and avoid repetitive use of tables. With these changes implemented, we hope that our paper is accepted for publication.

---

## [Editor Report · Decision Letter 2]

12 Apr 2024

Associations of body adiposity index, body mass index, waist circumference, and percentage of body fat in young female Emirati adults

PONE-D-23-35965R2

Dear Dr. Haroun,

We’re pleased to inform you that your manuscript has been judged scientifically suitable for publication and will be formally accepted for publication once it meets all outstanding technical requirements.

Kind regards,

Fredirick Lazaro mashili, MD, PhD

Academic Editor

PLOS ONE

Additional Editor Comments (optional):

The reviewer's concern regarding the manuscript presentation has been sufficiently addressed. All other comments raised by the reviewers have also been adequately addressed.